# Size, Value and Business Cycle Variables. The Three-Factor Model and Future Economic Growth: Evidence from an Emerging Market

**Fahad Ali [1],\* , RongRong He [2] and YueXiang Jiang [1]**

[1]  College of Economics, Zhejiang University, Hangzhou 310027, China; jiangyuexiang@zju.edu.cn
[2]  School of Business, Hong Kong Baptist University, Kowloon 999077, Hong Kong, China; lizzyhe@life.hkbu.edu.hk
\*  Correspondence: fahadali@zju.edu.cn; Tel.: +86-132-5081-2926

**Abstract:** The paper empirically investigates three different methods to construct factors and identifies some pitfalls that arise in the application of Fama-French's three-factor model to the Pakistani stock returns. We find that the special features in Pakistan significantly affect size and value factors and also influence the explanatory power of the three-factor model. Additionally, the paper examines the ability of the three factors to predict the future growth of Pakistan's economy. Using monthly data of both financial and non-financial companies between 2002 and 2016, the article empirically investigates and finds that: (1) size and book-to-market factors exist in the Pakistani stock market, two mimic portfolios SMB and HML generate a return of 9.15% and 12.27% per annum, respectively; (2) adding SMB and HML factors into the model meaningfully increases the explanatory power of the model; and (3) the model's factors, except for value factor, predict future gross domestic product (GDP) growth of Pakistan and remain robust. Our results are robust across sub-periods, risk regimes, and under three different methods of constructing the factors.

**Keywords:** emerging markets; KSE Pakistan; three-factor model; size and value premiums; future economic growth

## 1. Introduction

Asset pricing models are used to evaluate risk and return structure of stocks, and facilitate individual investors and institutions in planning and managing their portfolios. A list of models is available to assist investors and financial managers in predicting the expected return for their targeted stocks. However, two models are alternatively and widely used for this purpose. The first one is the Capital Asset Pricing Model (CAPM), developed by Sharpe (1964), Lintner (1965) and Mossin (1966). The second one is the three-factor model proposed by Fama and French (1993).

CAPM measures the sensitivity by a single factor: security's beta coefficient with a mean-variance coefficient of the market portfolio. In the early 1970s, CAPM, with its single factor to measure risk for a security, was widely used to facilitate the investors. Later, studies related to intertemporal asset pricing models (Merton 1973; Breeden 1979), arbitrage pricing theory (Ross 1976), size effect (Banz 1981; Reinganum 1981; Keim 1983), and book to market (value) effect (Rosenberg et al. 1985) and (Chan et al. 1991), highlighted some other variables having considerable effect on the relationship between average returns and systematic risk, that had remained unconsidered employing the single factor model, CAPM. These contributions helped to identify systematic risk empirically, which would not have been possible with the former abstracted and theoretical model.

The empirical validity of CAPM was struck down by the Fama and French (1992), which suggested that beta (β) is unable to fully capture the variations in cross-sectional expected returns. Later on, size (SMB) and book-to-market (HML) factors were also added as an extension of the CAPM (Fama and French 1993). It has been discussed that the three-factor model provides a better explanation as compared to CAPM in many countries.

Recently, Fama and French (2015) proposed a five-factor model which added investment (CMA) and profitability (RMW) as new factors into the existing three-factor model. However, the model is reportedly unable to explain the low average returns on small stocks, the returns of which are similar to those of firms with low profitability but high investment level. Elliot et al. (2016) discussed that such stocks are only a small fraction of the US market, but the case is different across global markets. They demonstrated that the newly added factors have limited explanatory power for these stocks. Moreover, the contribution of the value factor has been significantly diluted by the two new factors in terms of explaining the average returns; therefore, it has been suggested that the three-factor model generally fits well with Shanghai stock exchange (SSE) A-share market (Xie and Qu 2016).

The economic rationale (risk-based interpretation) behind the newly added factors (i.e., investment and profitability factors) has been criticized in recent studies (Ülkü 2017) and (Ali and Ülkü 2018). It has been explained that factors generally represent risk attributes (e.g., SMB and HM), while CMA and RMW are derived from the dividend discount model. Further, these factors capture mispricing away from 'value', caused by noise trading and the weekend sound-mind effect. Kubota and Takehara (2017) examined the five-factor model for the Japanese market. They concluded that the Fama-French's five-factor model is not the best benchmark for stocks traded at the Japanese stock market. Thus, the paper focuses on the three factors; size premium, value premium and the market risk premium.

In the context of Pakistan, Iqbal and Brooks (2007) analyzed the Fama-French three-factor model and CAPM for the stocks traded at Karachi stock exchange (KSE). They discussed that risk factors in the three-factor model are more significant. Mirza and Shahid (2008) deployed a multivariate framework to test the validity of the three-factor model. They included the stocks of financial firms as well; the results generally supported the three-factor model.

The Fama-French three-factor model has been widely applied to most of the developed and emerging markets; however, the model has been least applied to the emerging south-Asian market, Pakistan. This might be due to the small size of the market and the difficulty in assembling enough stocks to construct the underlying portfolios of this model in earlier decades. For instance, prior studies on KSE such as Iqbal and Brooks (2007), Mirza and Shahid (2008) and Javid and Ahmad (2008), which used a fixed number of stocks, ranging between 49 and 90 stocks, may have suffered from small sample problems. Moreover, the sample period in these studies either includes only bull rally between 2003 and 2007 or includes Asian-crises (1997), political instability in Pakistan (1999) and US-Afghan war (9/11); due to these events investment behaviors remained alert and risk averse. Hence, their results cannot be considered robust.

Fama-French excluded financial firms from their series of studies. They stated that "the stocks of financial firms are thinly traded and the financial firms tend to have higher financial leverage, but for non-financial firms, high leverage has a different meaning and can be considered as financial distress" (Fama and French 1992). Most of the studies followed the same approach and excluded the stocks of financial firms while empirically testing the three-factor model on various stock markets.

Pakistan shares the 'typical characteristics' and features of an emerging market, such as thick tails accompanied with excess kurtosis in the return distribution, high return with excessive volatility, and low market capitalization but high trading volume (Khwaja and Mian 2005). The special characteristics of financial firms in Pakistan, such as liquidity, active participation, and a large fraction of the market value of these firms to total market value of the index are rarely discussed in the literature. These characteristics are different as compared to the US and other developed stock markets discussed in Fama and French (1998, 2017), where stocks of financial firms are thinly traded and do not make up a large fraction of the total market value of the index.

Modigliani and Miller (1958, 1963) explain in theoretical language that the risk profile (beta) of the firm can be affected by leverage but it does not invalidate the fundamental principal of the asset pricing model. Therefore, it is better if the pricing model is generally applied, rather than restricted to nonfinancial firms only. Motivated by the Modigliani-Miller theory, Baek and Bilson (2015) assessed the size and value factors to measure the cross-section of expected stock return in financial and non-financial firms of US stock market. The empirical results suggested that size and value premiums commonly exist in both financial and non-financial firms. Therefore, we include both (financial and nonfinancial firms), as we believe that exclusion of financial sector firms is not justified in the case of Pakistan.

For the application of the three-factor model, factors formation methodology plays the most important role. In this regard, Xu and Zhang (2014) document the empirical evidence and identify some drawbacks that arise in the application of the three-factor model to Chinese stock returns. In order to evaluate the effect of several special features in China, they experiment with different ways to construct the three factors. Their results illustrate that the construction of the three factors can have a significant impact in empirical studies that apply the Fama-French's three-factor model. Further, Vo (2015) and Xie and Qu (2016) also discussed the applicability of the three-factor model by considering the special features of the Australian stock market and the SSE (A shares) China, respectively.

However, in the context of the Pakistani stock market, it is still unclear whether the portfolio construction should be based on Fama-French (i.e., to exclude the stocks of financial firms), or on adapting the strategy of keeping fixed number of stocks, similar to the previous studies on KSE, or on including stocks of both financial and non-financial companies as well as new firms.

To adjust for the small sample problem, a unique feature of KSE and the investment behaviors across the diverse economic conditions, this article uses a larger dataset (2002–2015) as compared to any previous study on KSE. A comparatively larger dataset containing all the liquid stocks to avoid the illiquidity factor (zero returns), is expected to improve the power of the tests and will capture variation in stock returns beyond any previous study on KSE. The paper argues for the importance of the special features in the Pakistani market and compares three different factors construction methodologies which may significantly affect the performance of the three-factor model. The three different constructed baskets of stocks are: 'fixed basket', 'non-financial basket' and 'variable basket'. The fixed basket includes only those stocks which survive the entire sample period, the non-financial basket and variable basket include (exclude) companies into (from) the basket every year upon meeting (differing from) the sample selection and criteria limitations. The variable basket includes all the stocks, whilst the non-financial basket only includes non-financial stocks.[1]

The summary statistics, reported in Table 1, confirm that the monthly returns on the factor portfolios in three different scenarios are somewhat different from each other. For example, the fixed basket generates approximately 5.66% per annum size premium, whereas the non-financial and variable baskets generate approximately 9.14% and 9.15% per annum, respectively. The value premium for the fixed basket is approximately 11.25% per annum, whereas the non-financial and variable baskets generate approximately 15.04% and 12.27% per annum, respectively. The significance of these factors also varies. The value premium is significant at a 5% level in each of the portfolio construction methodologies. Conversely, SMB is statistically insignificant for the fixed basket, significant at 10% level for the non-financial basket, and significant at 5% level for the variable basket.

---

[1]　See Section 3.4 of this paper for further details.

**Table 1.** Descriptive statistics: comparing three different methods of constructing factors.

| | Fixed Basket | | Non-Financial Basket | | Variable Basket | |
|---|---|---|---|---|---|---|
| | SMB | HML | SMB | HML | SMB | HML |
| Mean (%) | 0.471 | 0.938 | 0.762 | 1.254 | 0.763 | 1.022 |
| STD (%) | 5.252 | 6.197 | 5.759 | 7.017 | 5.323 | 6.529 |
| *t*-statistic | 1.163 | 1.961 ** | 1.715 *** | 2.315 ** | 1.858 ** | 2.029 ** |

Note: Authors calculation. The table reports the comparison of descriptive statistics of size (SMB) and value (HML) premiums between the fixed basket, non-financial basket and variable basket. The sample period is 2002:01–2015:12, *** and ** indicate significance at 10 and 5% level, respectively. Source: the official website of the Pakistan stock exchange (https://www.psx.com.pk/) and the official website of the State Bank of Pakistan (http://sbp.org.pk/).

Liew and Vassalou (2000) analyzed the relationship between future economic growth and the Fama–French three-factor model. Their findings suggest that size and book-to-market factors are positively related to future economic growth. Vassalou (2003) and Petkova (2006) observed a moderated explanatory power of the Fama–French factors in the existence of macro-economic risk. The findings by Boamah (2015) provide a further indication of the relevance of SMB and HML to future economic growth.

There is no known study that has observed the predictive ability of these risk factors for future economic growth in the GDP of the Pakistani economy. However, Javid and Ahmad (2008) examine a set of macroeconomic variables in addition to market risk premium (single factor). The results support the proposal that a few economic variables play an additional role in explaining the variation in stock returns and this variability has some business cycle correlations.

The paper offers several pioneering contributions. It: (1) constructs and compares the risk factors and the three-factor model under three different methods; (2) examines the robustness of the model across different risk regimes and subsamples; (3) analyses the performance of the term structure premium augmented four-factor model; and (4) links the information content of the Fama–French factors and the business cycle variables to predict future economic growth of Pakistan. The evidence will enhance our understanding of whether or not these factors relate to underlying economic risk factors.

Our results show that: (1) size and value premiums exist in the KSE. In terms of returns, small size firms outperform big size firms while value stocks (high book-to market equity ratio) outperform growth stocks (low book-to-market equity ratio); (2) the three-factor model can explain the variations in average stock returns on six size-B/M portfolios, the average adjusted R squared meaningfully increased by including two additional factors into the model; (3) the three-factor model by constructing portfolios in different ways, is applicable to KSE, as all the models capture size and book-to-market effects significantly; (4) the significance of the regression coefficients are time variant. However, the existence of these factors is stable across the three sub-periods; (5) the three-factor model captures the size and book-to-market effects significantly for the six risk-based (categorized based on market beta) portfolios; (6) loadings on the term structure premium (TSP) mostly remain statistically significant but do not improve the explanatory power of the augmented four-factor model, contrarily it increases the significance of the intercept. However, SMB and HML remain robust in the presence of the TSP; and (7) the market and SMB factors possess the predictive ability for one-year ahead growth of the Pakistani economy and remain robust in the presence of the business cycle variables.

This paper proceeds as follows. Section 2 provides a related literature review of prior studies, Section 3 describes the data and the methodologies used in the paper. Section 4 discusses empirical results and analysis. Section 5 investigates the relevance of the risk factors to predict future economic growth. Section 6 summarizes the research findings and concludes the paper.

## 2. Prior Related Studies

Success of the Fama-French three factor model is, basically, a divergence in CAPM and emerged as a most popular explanation for the ongoing argument on asset pricing. However, several studies in the financial literature (e.g., Groenewold Fraser 1997; Beltratti and Tria 2002; Drew and Veeraraghan 2002; Mirza and Shahid 2008; Guo et al. 2008; Lischewski and Voronkova 2012; Cakici et al. 2013; Minović and Živković 2014; Baek and Bilson 2015; Boamah 2015; Ceylan et al. 2015; Zaremba and Konieczka 2015; Elgammal et al. 2016; Chung et al. 2016; Xie and Qu 2016; Kubota and Takehara 2017) attribute mixed evidence regarding the existence, significance, augmented versions and time varying behavior of the risk premiums and the three-factor model in the stock markets of USA, Europe, Australia, Asia and Africa by applying various models and portfolio construction methodologies.

Daniel and Titman (1997) examined the Fama and French (1993) and demonstrated that size and book-to-market factors are highly correlated with the average stocks returns but there is no separate distress and most of the co-movement of the value stocks is not due to distressed stocks being exposed to a unique distress factor. They explained that it is characteristics rather than factor loadings that appear to explain the cross-sectional variation in stock returns. Davis et al. (2000) thoroughly studied the characteristics, co-variances and average returns for the period (1929–1997). By dividing the sample into two sub-periods, their findings confirmed that value premium (HML) factor was 0.50% per month in the first sub-period (1929–1963) and 0.43% per month in the second sub-period (1963–1997). The Value premium observed in the first sub-period was statistically significant at ($t$ = 2.8) while the second sub-period presented higher significance at ($t$ = 3.38). They confirmed a strong relationship between value premium and average stock returns. They discovered that the results of Daniel and Titman (1997) appeared to be supporting characteristics of the model due to the shorter time span.

Connor and Sehgal (2001) analyzed the results of the Fama-French three-factor model and CAPM. Stocks traded at the CRISIL 500 Indian stock market were taken as a sample. The results after using wald statistics showed that three out of six portfolios had significant intercepts for CAPM, whereas, in the Fama-French model all six portfolios had insignificant intercepts. Finally, on the basis of their findings, it was concluded that the three-factor model performs better for the Indian stock market than the CAPM.

In their study of three developed markets, Griffin (2002) found that the three-factor model can significantly explain the variations in the cross-section of expected stock returns in the stock markets of Canada, England and Japan. Drew and Veeraraghan (2002) detected size and value premiums in the Malaysian stock market. De Groot and Verschoor (2002) analyzed the influence of size and value factors on stocks' average returns in five Asian emerging markets. Their findings suggested a strong size effect for all of the markets (India, Korea, Malaysia, Taiwan and Thailand), while value effect only exists in Thailand, Malaysia and Korea.

For the Australian stock market, O' Brien et al. (2008) compared the CAPM with the three-factor model. Their results suggested that the three-factor model explained nearly 70% of the variations in return and led to the formation of an opinion that the three-factor model is a very effective and useful model for explaining the variation in expected stock returns. Brown et al. (2008) detected time-varying value premium in the stock markets of Hong Kong, Korea and Singapore. However, they found a value discount in the Taiwanese stock market.

Malkiel and Jun (2009) studied the Chinese stocks and confirmed the existence of size and book-to-market effects for returns on Chinese stocks. Lischewski and Voronkova (2012) examined the factors determining the stock prices on the Polish stock market (WSE). Findings supported the existence of the size and value factors along with the market risk premium, while liquidity factor was not priced in Polish stocks.

Xu and Zhang (2014) empirically investigated the Fama-French three-factor model and identified some downsides that can arise in the application of the three-factor model to the Chinese stock returns. In order to evaluate the effect of several special features in China, they experiment with different ways to construct the three factors. They concluded that formation of the three factors can have a significant

impact in empirical studies that apply the three-factor model to Chinese stock market. In the same way, Vo (2015) examined various approaches to construct portfolios and proposed further evidence for the Australian market.

Therefore, Xie and Qu (2016) performed an empirical study, by focusing on the unique features of the Chinese stock market. Their study consisted of stocks traded at SSE A-share between 2005 and 2012. The findings suggested that size and value premiums are significant for China's stock market (SSE A-share market) and the three-factor model generally fits well. They did not include the investment and profitability factors in the model as these factors dilute the value factor. Similarly, Kubota and Takehara (2017) empirically investigated and rejected the Fama-French's five-factor model as a benchmark for the Japanese stock market.

In a Pakistani context, Iqbal and Brooks (2007) analyzed the conditional Fama-French three-factor model and CAPM for the stocks traded at KSE-Pakistan. The GARCH and EGARCH methods are used on monthly, weekly and daily data of 89 stocks during the period between 1992 and 2006. They illustrated in a graphical analysis that conditioning variables generally result in upward bias. They concluded that the unconditional three-factor model performs better. Mirza and Shahid (2008) deployed a multivariate framework to test the validity of the three-factor model. The sample consisted of 81 stocks traded on KSE from January 2003 to December 2007. The results confirmed the size premium but reported a value discount. Their findings, in general, supported the three-factor model. Javid and Ahmad (2008) examined a set of macroeconomic variables along with the market risk premium on 49 stocks traded at KSE during the period between 1993 and 2004. The results supported that the economic variables play an incremental part in explaining the variation in stock returns and this variability has some business cycle correlations.

Within a broad international analysis Liew and Vassalou (2000) examined the relationship between the Fama–French's three factors and future economic growth in ten countries. The results indicated that SMB and HML are positively related to future economic growth. The predictive ability of the Fama–French factors is found independent on the market factor. They contended that their findings support the risk-based interpretation of the Fama–French factors. Further, a moderate explanatory power of the Fama–French factors for stock returns in the presence of macroeconomic risk factors is noticed by several studies (Aleati et al. 2000; Lettau and Ludvigson 2001; Vassalou 2003; Petkova 2006). Similarly, Boamah (2015) examined the applicability of the Fama-French factors and explore the ability of these factors to predict future economic growth (GDP) of South Africa. The findings show the relevance of small firms and value stocks on the South-African stock market. Additionally, the results show a significantly positive relationship between future economic growth and SMB, HML, and the market factor. The findings remain robust to the inclusion of business cycle variables in the model.

## 3. Data and Methodology

Emerging markets have their own dynamics, significantly different from developed markets (Bruner et al. 2002). KSE was declared as an open market in 1991 but the pace of the market was stagnant until 2001. However, the market has shown a tremendous growth in recent years; the index has grown by more than 715% in the last eight years (December 2008 to December 2016). Our dataset consists of stocks traded at KSE from January 2002 to December 2015 and GDP growth rates between 2003 and 2016.[2] We start from January 2002 due to a number of reasons, such as: (1) the availably of the data on the official website of the KSE; (2) the stocks remain actively traded at KSE in this period; and (3) in preceding years, the market was illiquid and influenced by other global and regional

---

[2] Pakistan has three stock markets, the other two stock markets are Islamabad stock exchange and Lahore stock exchange, however all these three markets were merged on 11 January 2016 and renamed as Pakistan stock exchange. Source: https://www.psx.com.pk/.

factors.[3] Thus, it is better to include a lag of a few months to avoid potential bias and begin taking data from January 2002. The study which spans the period of 168 months, including bearish, bull, super bull, recession, recovery and, again, rapid growth in the market, covers all characteristics of market performance and is long enough to ensure stability and efficacy of the model.

*3.1. Types and Sources of Data*

Data on stock prices and index closing points are obtained from the official website of KSE.[4] The cut-off yield on the Pakistani Treasury bill rate (T-bill), Pakistan investment bonds (PIBs), and financial statements of financial sector data are obtained from the official website of the State Bank of Pakistan (SBP).[5] The financial daily Business Recorder is used for the data related to number of outstanding shares, market capitalizations and any other missing information.[6] The KSE-100 index is a market capitalization weighted index and is used as the market return, whereas 6 month T-bills cut-off yields are converted into monthly values and used as a risk-free rate, similar to the previous studies on KSE (Iqbal and Brooks 2007; Mirza and Shahid 2008). Overall, more than 630 stocks are carefully observed. However, after screening the stocks as per criteria limitations, the number of stocks included are reduced to 330.[7] Table 2 shows the number of stocks considered in each case. A continuous change in the number of stocks can be noticed across different baskets and this may present different results. We include delisted firms in the sample up to the delisting year to control the survivorship bias. The dataset is modified on December 31 each year. In order to estimate the monthly returns, the closing price of the last day of each month is used.

**Table 2.** Year-over-year sample size (2002 to 2015).

| Year | Fixed Basket | Non-Financial Basket | Variable Basket |
|------|------|------|------|
| 2002 | 192 | 162 | 195 |
| 2003 | 192 | 168 | 202 |
| 2004 | 192 | 188 | 226 |
| 2005 | 192 | 200 | 249 |
| 2006 | 192 | 210 | 258 |
| 2007 | 192 | 212 | 269 |
| 2008 | 192 | 214 | 277 |
| 2009 | 192 | 234 | 305 |
| 2010 | 192 | 243 | 313 |
| 2011 | 192 | 248 | 320 |
| 2012 | 192 | 253 | 326 |
| 2013 | 192 | 254 | 327 |
| 2014 | 192 | 257 | 330 |
| 2015 | 192 | 257 | 330 |

Note: Author's calculation. The table reports the number of companies considered for the reformation of six size and book-to-market (B/M) sorted portfolios each year-end from 2002 to 2015.

*3.2. Selection Criteria and Limitations*

For selected companies, monthly price data, market value of equity, book value and other fundamental information should be available; selected stock must survive for a complete year and be traded for at least 85% of the trading days with non-zero returns during the year.

---

3    These factors include, but are not limited to Asian crises (1997), political uncertainty in Pakistan (1999) and US-Afghan (9/11) war.

4    The official website of Karachi stock exchange is www.kse.com.pk (new: https://www.psx.com.pk/).

5    The official website of the State Bank of Pakistan (SBP) is www.sbp.org.pk.

6    Source: http://www.brecorder.com/market-data/karachi-stocks/.

7    See, Section 3.2 for selection criteria and limitations.

### 3.3. Model Specification

In order to test the significance and existence of the diverse factors on asset pricing in the Pakistani stock market (KSE), we employ numerous pricing models and follow a stepwise approach. We start with a standard CAPM:

$$E(R_i) - R_f = \alpha_i + \beta_i\left[E(R_m) - R_f\right] + \epsilon_i \tag{1}$$

Next, we add SMBL and HML factors into the CAPM:

$$E(R_i) - R_f = \alpha_i + \beta_i\left[E(R_m) - R_f\right] + s_i(SMB) + h_i(HML) + \epsilon_i \tag{2}$$

where, $E(R_i) - R_f$ is the portfolio $i's$ return in excess of risk-free rate $R_f$, $\alpha_i$ is the intercept of the regression equation representing the non-market return component, $E(R_m) - R_f$ is the market risk premium (market portfolio return in excess of risk-free rate), SMB (small minus big) is the return on small size stocks minus return on big size stocks captures size premium, HML (high minus low) incorporates value premium that is the difference between returns of value stocks (high B/M ratio) and growth stocks (low B/M ratio). $\beta_i$, $s_i$ and $h_i$, are the slopes of expected risk premium of portfolio $i$ to the market, size and value factors in the regression, respectively, while $\epsilon_i$ represents the random return component due to unexpected events related to a particular portfolio. It is supposed that $\epsilon_i$ has a multivariate normal distribution and is identically and independently distributed over time.

### 3.4. Variable Construction and Portfolio Formation

In order to examine the three factors for Pakistani stocks, we experiment with three ways of constructing size and value factors to explore the impact of the special features in the Pakistani stock market. The three portfolio construction methods (baskets of stocks) are: 'fixed basket', 'non-financial basket' and 'variable basket'. By following the previous studies on KSE, we construct the fixed basket; which includes only those stocks which have survived the entire sample period. Next, we follow Fama and French (1993) and construct the non-financial basket. The non-financial basket excludes stocks of financial companies, however, every year new companies are included into the basket upon meeting the sample selection and criteria limitations. The variable basket is based on the special features of KSE, such as: liquidity of the financial companies, active participation, and fraction of the market value of the financial firms to the total market value of the index. It includes both non-financial and financial companies into the basket every year upon meeting the sample selection and criteria limitations. The variable and the non-financial baskets include delisted firms in the sample up to the delisting year to control the survivorship bias, whilst the fixed basket does not include.

The dependent variable of the three factor model is the excess return on equal weighted six portfolios. 2 pcs (small and big) portfolios are determined for size effect and 3 pcs (high, medium, low) portfolios are determined for value effect. A total of six intersection portfolios (SL, SM, SH, BL, BM, BH) are created with the following criteria: shares classified according to market value have been subdivided using the median market cap as breakpoint, while shares classified according to book-to-market have been divided by the 30th and 70th percentiles as breakpoints. In the case of risk-based portfolios, i.e., constructing portfolio by categorizing the stock's sensitivity to market movements, six portfolios on the basis of risk profile of the stocks are carefully considered as the dependent variables.

The independent variables consist of market, size and value premiums. For the market risk premium, we find the difference between the return on market portfolio and risk free rate, and show that it exists in both Fama-French three factor model and CAPM. Size premium (SMB) is the average return on three small portfolios SL, SM and SH minus the average return on the three big portfolios BL,

BM and BH, while HML is the average return on two value portfolios SH and BH minus the average return on the two growth portfolios SL and BL. SMB and HML are computed as follows:

$$\text{SMB} = \frac{(\text{SL} + \text{SM} + \text{SH})}{3} - \frac{(\text{BL} + \text{BM} + \text{BH})}{3} \tag{3}$$

$$\text{HML} = \frac{(\text{SH} + \text{BH})}{2} + \frac{(\text{SL} + \text{BL})}{2} \tag{4}$$

## 4. Empirical Results and Discussion

The descriptive statistics in Table 1 report that the average monthly returns on the SMB are statistically insignificant in the fixed basket, and significant at 10% and 5% levels in non-financial and variable baskets, respectively. In contrast, value premium is significant at 5% level across all the methodologies. To understand the changes in the results caused by construction methodologies, we start with a detailed analysis of the variable basket, because it is significant for both factors at 5% level. Afterwards, we compare the performance of these factors obtained by three different portfolio construction methodologies. Table 3 reports the descriptive statistics of the monthly excess return and volatility of the six size-B/M sorted portfolios from January 2002 to December 2015 obtained by using variable basket.

**Table 3.** Descriptive statistics on the excess return (and volatility).

|  | Low B/M | Medium B/M | High B/M |
| --- | --- | --- | --- |
| Small Capitalization | 1.384 (9.008) | 1.689 (7.193) | 3.462 (10.375) |
| Big Capitalization | 1.427 (7.297) | 1.426 (8.285) | 1.394 (11.150) |

Note: Author's calculation. The table reports the descriptive statistics on monthly average excess returns between six size-B/M portfolios for the Pakistani stock market. The values of standard deviation are reported in parentheses. The sample period is 2002:01–2015:12 (168 monthly observations). Source: the official website of the Pakistan stock exchange (https://www.psx.com.pk/) and the official website of the State Bank of Pakistan (http://sbp.org.pk/).

Holding group size constant, the average return and volatility of the portfolios increase with the portfolio's B/M ratio. The average monthly return on portfolios containing low B/M is 1.41% and the standard deviation is 8.15%, whereas stocks with high B/M ratio have an average return of 2.43% and a standard deviation of 10.76%. Conversely, when the B/M ratio is constant, the average return on small capitalization firms is 2.42% and the standard deviation is 9.69%, whereas stocks with big capitalization have an average return of 1.41% and a standard deviation 9.22%. Average returns on all portfolios are positive in our study, but are contradictory to the results reported by Mirza and Shahid (2008). Their results for KSE during the bull rally between 2003 and 2007 report negative average monthly returns on portfolios SH, BH and BM. The monthly average returns are the highest in the small value category (SH), approximately 3.46%, while the lowest in the small growth category (SL) is approximately 1.38%. The monthly standard deviation is the highest in the big value category (BH), approximately 11.15% and the lowest in the small medium-B/M category (SM), approximately 7.19%.

Table 4 represents the summary statistics of all three portfolios for time period from January 2002 to December 2015. The monthly average returns of the three explanatory variables are all positive and significant. The annual average return on the market, size and value factors is approximately 18.22%, 9.15% and 12.27%, respectively, whereas the standard deviation is approximately 7.60%, 5.32% and 6.53%, respectively. It is evident from the results that small stocks and values stocks outperform the big stocks and growth stocks, respectively.

**Table 4.** Summary statistics of independent variables (factors).

|  | $R_m - R_f$ | **SMB** | **HML** |
|---|---|---|---|
| Mean (%) | 1.518 | 0.763 | 1.022 |
| STD (%) | 7.596 | 5.323 | 6.529 |
| *t*-statistic | 2.591 | 1.858 | 2.029 |

Note: Author's calculation. The table reports the summary statistics of the market risk premium ($R_m - R_f$), size premium (SMB) and value premium (HML). The sample period is 2002:01–2015:12 (168 monthly observations). Source: the official website of the Pakistan stock exchange (https://www.psx.com.pk/) and the official website of the State Bank of Pakistan (http://sbp.org.pk/).

Table 5 reports the correlation coefficients among the independent variables. We did not notice any excessively high values of the correlation coefficients that may arise a concern about any multicollinearity problem. The observed correlation shows that SMB and HML can be regarded as separate measures of risk premium, which are not dependent on market risk premium. The correlation between SMB and HML also shows a valid justification for considering size and value risk factors separately.

**Table 5.** Correlation coefficients of monthly factor returns.

|  | $R_m - R_f$ | **SMB** | **HML** |
|---|---|---|---|
| $R_m - R_f$ | 1 |  |  |
| SMB | −0.445 | 1 |  |
| HML | 0.365 | −0.176 | 1 |

Note: Author's calculation. The table reports the correlation coefficients between market, size and value factors. The sample period is 2002:01–2015:12 (168 monthly observations). Source: the official website of the Pakistan stock exchange (https://www.psx.com.pk/) and the official website of the State Bank of Pakistan (http://sbp.org.pk/).

*4.1. Regression Results*

In this section, we analyze the standard CAPM and the Fama-French three-factor model by employing time-series regression for each of the six size-B/M portfolios (SL, SM, SH, BL, BM, BH). The objective of this approach is to identify the role of size and value factors to capture variation in stock returns during the period from January 2002 to December 2015. We start from the traditional single factor CAPM in Table 6, in order to make a comparison with the three-factor model later.

**Table 6.** Capital Asset Pricing Model (CAPM) regressions on monthly excess returns of portfolios formed on size and B/M ratio (variable basket).

| $R_i - R_f$ | **α** | **β** | $R^2$ | **Adj. $R^2$** |
|---|---|---|---|---|
| SL | 0.006 (0.915) | 0.528 (6.406) * | 0.198 | 0.193 |
| SM | 0.008 (1.778) *** | 0.597 (10.454) * | 0.397 | 0.393 |
| SH | 0.023 (3.387) * | 0.786 (9.070) * | 0.331 | 0.327 |
| BL | 0.002 (0.682) | 0.795 (18.971) * | 0.684 | 0.682 |
| BM | 0.001 (0.208) | 0.887 (18.014) * | 0.662 | 0.659 |
| BH | −0.004 (−0.698) | 1.165 (16.789) * | 0.629 | 0.627 |

Note: Author's calculation. The table reports the estimation results of the single factor CAPM. Stocks are sorted into six size-B/M portfolios (SL, SM, SH, BL, BM, BH). *t*-stats are in parenthesis, *** and * indicate significance at 10% and 1% level, respectively. The sample period is 2002:01–2015:12 (168 monthly observations). Source: the official website of the Pakistan stock exchange (https://www.psx.com.pk/) and the official website of the State Bank of Pakistan (http://sbp.org.pk/).

The results reported in Table 6 show that the average adjusted $R^2$ value of the CAPM is approximately 48%, suggesting that the CAPM does not explain most of the time-series variations in stock returns. The intercept of the CAPM is statistically significant for two out of six portfolios, i.e., small stocks with medium B/M ratio (SM) and small value stocks (SH). The portfolios containing small stocks generate higher average intercept and lower $R^2$. Thus, the results of the CAPM regressions provide some preliminary indication for a size premium.

Next, we include size and book-to-market factors into the model. Table 7 reports the results of the Fama-French three-factor model based on variable basket. The $R^2$ of the six regressions, with an average of approximately 71.74%, are much higher than those of the CAPM regressions. Usually, adding an independent factor into regression increases $R^2$. If the change is meaningfully higher, it is considered to be an improvement in the model. The average value of $R^2$ for small size group increases from approximately 30.88% to 71.08%, signifying that the three-factor model provides a massive improvement in the explanatory power over the CAPM. Therefore, our regression results support the argument that the three-factor model is a much better fit for KSE, Pakistan.

**Table 7.** Three factor regression on monthly excess returns of portfolios formed on size and B/M ratio (variable basket).

| $R_i - R_f$ | $\alpha$ | $\beta$ | s | h | $R^2$ | Adj. $R^2$ |
|---|---|---|---|---|---|---|
| SL | −0.007 (−1.599) | 0.988 (14.889) * | 1.182 (13.197) * | −0.291 (−4.149) * | 0.632 | 0.625 |
| SM | −0.002 (−0.695) | 0.762 (15.023) * | 0.737 (10.764) * | 0.205 (3.821) * | 0.663 | 0.657 |
| SH | 0.004 (1.067) | 0.906 (17.861) * | 1.187 (17.330) * | 0.796 (14.824) * | 0.838 | 0.835 |
| BL | 0.003 (0.895) | 0.860 (17.961) * | 0.025 (0.381) | −0.183 (−3.615) * | 0.708 | 0.703 |
| BM | −0.001 (−0.220) | 0.855 (14.952) * | 0.061 (0.788) | 0.164 (2.706) * | 0.677 | 0.671 |
| BH | −0.008 (−1.882) *** | 0.942 (15.082) * | 0.020 (0.235) | 0.729 (11.036) * | 0.787 | 0.783 |

Note: Author's calculation. The table reports the estimation results of the three-factor model (variable basket). Stocks are sorted into six size-B/M portfolios (SL, SM, SH, BL, BM, BH). *t*-Stats are in parenthesis, *** and * indicate significance at 10% and 1% level, respectively. The sample period is 2002:01–2015:12 (168 monthly observations). Source: the official website of the Pakistan stock exchange (https://www.psx.com.pk/) and the official website of the State Bank of Pakistan (http://sbp.org.pk/).

Theoretically, if a model satisfactorily explains the changes in the expected returns, then the intercept produced by regression results will tend towards zero. Table 7 reports that the six size-B/M portfolios produce intercepts, ranging from −0.0072 to 0.0037, are close to zero. Only the portfolio containing stocks with a high market capitalization and a high book-to-market value (BH) shows a significant (negative) intercept. The significant intercept for portfolio BH indicates that the big value firms have something not predicted by the model.

The loadings on the market factor are all significant at 1% level and thus reflect a positive sensitivity to market risk. HML has significantly positive coefficients for high B/M firms and significantly negative for low B/M firms. The coefficients of value stocks have both a large and positive sensitivity to HML, whereas growth stocks have a low and negative sensitivity to HML. Our results support the existence of value premium. Similarly, the coefficients of small firms have both a large and positive sensitivity to SMB, whereas the big firms have insignificant sensitivity. The coefficients of big firms are very small, ranging from approximately 0.019 to 0.061, whereas the coefficients of small firms range from approximately 0.737 to 1.187. Although the insignificant sensitivity of big firms to SMB is different from Fama-French's findings, the coefficient of small firms

are highly significant both in the economic and statistically sense. This finding also indicates adequate evidence to support the existence of a size premium.

### 4.2. Comparative Analysis of the Three-Factor Model

In this section, we examine the three-factor model based on 'fixed' and 'non-financial' baskets. For comparison, Table 8 represents the three factor model regression results.

**Table 8.** Three factor regression on monthly excess returns of portfolios formed on size and B/M ratio (fixed basket and non-financial basket).

| $R_i - R_f$ | $\alpha$ | $\beta$ | s | h | $R^2$ | Adj. $R^2$ |
|---|---|---|---|---|---|---|
| **Panel A: Fixed basket** | | | | | | |
| SL | −0.002 (−0.341) | 0.937 (14.412) * | 1.286 (14.043) * | −0.442 (−5.945) * | 0.657 | 0.651 |
| SM | 0.001 (0.397) | 0.703 (13.838) * | 0.704 (9.829) * | 0.100 (1.717) *** | 0.591 | 0.584 |
| SH | 0.007 (1.708) *** | 0.876 (16.413) * | 1.175 (15.623) * | 0.758 (12.428) * | 0.789 | 0.785 |
| BL | 0.007 (1.826) *** | 0.789 (15.741) * | −0.003 (−0.044) | −0.240 (−4.180) * | 0.635 | 0.629 |
| BM | 0.001 (0.343) | 0.877 (15.481) * | 0.060 (0.747) | 0.095 (1.468) | 0.648 | 0.641 |
| BH | −0.002 (−0.355) | 0.850 (13.821) * | 0.108 (1.250) | 0.561 (7.976) * | 0.683 | 0.677 |
| **Panel B: Non-financial basket** | | | | | | |
| SL | −0.003 (−0.509) | 1.076 (14.413) * | 1.231 (13.013) * | −0.348 (−4.502) * | 0.643 | 0.636 |
| SM | 0.000 (0.035) | 0.690 (13.701) * | 0.717 (11.221) * | 0.135 (2.580) * | 0.626 | 0.619 |
| SH | 0.008 (2.024) ** | 0.842 (15.513) * | 1.145 (16.654) * | 0.807 (14.367) * | 0.813 | 0.810 |
| BL | 0.008 (1.997) ** | 0.748 (14.334) * | −0.007 (−0.103) | −0.200 (−3.705) * | 0.582 | 0.574 |
| BM | 0.010 (0.220) | 0.877 (15.0450) * | 0.021 (0.280) | 0.150 (2.480) ** | 0.653 | 0.647 |
| BH | −0.003 (−0.635) | 0.983 (14.338) * | 0.079 (0.911) | 0.645 (9.083) * | 0.724 | 0.719 |

Note: Author's calculation. The table reports the estimation results of the three-factor model (fixed and non-financial basket). Stocks are sorted into six size-B/M portfolios (SL, SM, SH, BL, BM, BH). *t*-stats are in parenthesis, ***, ** and * indicate significance at 10%, 5% and 1% level, respectively. The sample period is 2002:01–2015:12 (168 monthly observations). Source: the official website of the Pakistan stock exchange (https://www.psx.com.pk/) and the official website of the State Bank of Pakistan (http://sbp.org.pk/).

The average $R^2$ value of fixed basket and non-financial basket is 66.71% and 67.72%, respectively. Two portfolios, namely SH and BL, represent statistically significant intercepts for both of the baskets. The intercepts of SH and BL range from −0.0016 to 0.0065 and −0.0027 to 0.0079, respectively. The coefficients of the market factor are all positive and significant at 1% level. The coefficients of small portfolios have a large and positive sensitivity to SMB, whereas big portfolios have a smaller and positive sensitivity to SMB. Finally, the coefficients of high B/M portfolios have a large and positive sensitivity to HML factor, while the low B/M portfolios have a smaller and negative sensitivity to HML. Our regression results are mostly similar to the variable basket. Similarly, we analyze CAPM based on fixed and non-financial baskets, in order to make a reasonable comparison with the CAPM-variable

basket. However, the primary interest in the CAPM regression is the $R^2$ values and the intercept term. The results reported in Table A1 (Panel A and Panel B) in the appendix demonstrate that the fixed and non-financial baskets exhibit similar patterns as the variable basket. That is, the portfolios containing small stocks generate higher average intercept and lower $R^2$.

### 4.3. Model Performance Summary

One of our primary tasks is to test how well the three different types of factors construction explain average excess returns on the portfolios. Table 9 (Panel A and Panel B) examines the average absolute intercept (A$|\alpha_i|$), a measure of unexplained proportion of time series return variance ($1 - R^2$), and the number of portfolios with statistically significant intercept (NPSIs). Panel C of Table 9 estimates the mean, standard deviation, Sharpe ratio and cumulative wealth of size and value factors.

**Table 9.** Summary statistics for tests of CAPM, three-factor model, and size and value premiums.

| Factor Construction Type | A$\|\alpha_i\|$ | A $(1 - R^2)$ | NPSIs |
|---|---|---|---|
| **Panel A: CAPM** | | | |
| CAPM- fixed basket | 0.008 | 0.568 | 2 |
| CAPM- non-financial basket | 0.009 | 0.565 | 2 |
| CAPM- variable basket | 0.007 | 0.516 | 2 |
| **Panel B: Three-factor model** | | | |
| Fama-French- fixed basket | 0.003 | 0.333 | 2 |
| Fama-French- non-financial basket | 0.004 | 0.327 | 2 |
| Fama-French- variable basket | 0.004 | 0.283 | 1 |

| **Panel C: Size and value premiums** | | | | |
|---|---|---|---|---|
| Factors | Mean% | Std. Dev.% | Sharpe Ratio | Cumulative Wealth |
| SMB (FB) | 0.4714 | 5.252 | 0.089 | 1.792 |
| SMB (NB) | 0.7618 *** | 5.759 | 0.132 | 2.279 |
| SMB (VB) | 0.7628 ** | 5.323 | 0.143 | 2.282 |
| HML (FB) | 0.9375 ** | 6.197 | 0.151 | 2.575 |
| HML (NB) | 1.2535 ** | 7.017 | 0.179 | 3.106 |
| HML (VB) | 1.0222 ** | 6.529 | 0.157 | 2.717 |

Note: Author's calculation. Panel A and Panel B show the average absolute intercept A$|\alpha_i|$, the measure of unexplained proportion of time-series return variance ($1 - R^2$), and the number of portfolios with statistically significant intercepts NPSIs of CAPM and the three-factor model, respectively. Panel C reports the mean, standard deviation, Sharpe ratio and cumulative wealth of size and value factors based on three different portfolio construction methodologies. FB, NB and VB represent the fixed basket, non-financial basket and variable basket, respectively. *t*-stats are in parenthesis, *** and ** indicate significance at 10% and 5% level, respectively. The sample period is 2002:01–2015:12 (168 monthly observations). Source: the official website of the Pakistan stock exchange (https://www.psx.com.pk/) and the official website of the State Bank of Pakistan (http://sbp.org.pk/).

Panel A of Table 9 shows that the average absolute intercepts and the average unexplained portion of the time-series returns variance of CAPM are the lowest in the variable basket. Similarly, the average values of $(1 - R^2)$ across the 6 LHS portfolios, measuring the unexplained portion of the time-series return variance of the three-factor model are approximately 33.29%, 32.65% and 28.26% for fixed, non-financial and variable baskets, respectively. These findings confirm that the explanatory power of the variable basket for both the three-factor model and CAPM is higher than the other two methods of factors construction. Table 9 (Panel B) further validates that the average absolute intercepts and the average values of $(1 - R^2)$ of the three-factor model are generally smaller than those of the CAPM. The variable basket produces only one portfolio that generates statistical significant intercept, whereas the other two baskets produce two portfolios with significantly positive intercepts. Further, Panel C of Table 9 shows that the Sharpe ratio, cumulative wealth and the significance level (5%) of SMB are higher in variable basket than non-financial and fixed baskets. The HML is significant at 5% level in all

the three baskets, however the Sharpe ratio and cumulative wealth are higher if HML is constructed by including only non-financial stocks.

Figure 1 plots the cumulated monthly value of one rupee (nPKR) invested at the start of January 2002 and compounded at the monthly returns of the two factors (SMB and HML) in the KSE, Pakistan. The solid lines represent the two factors constructed by using fixed basket (SMBf and HMLf). The round-dotted lines represent the non-financial basket (SMBn and HMLn), and the square-dotted lines represent the variable basket (SMBv and HMLv). The time period is from January 2002 to December 2015. Figure 1 shows that both factors follow the same pattern under all the three ways of constructing the factors. The cumulative wealth of SMB factor is the highest when it is constructed based on the variable basket, whereas the cumulative wealth of HML factor is the highest when it is constructed based on the non-financial basket.

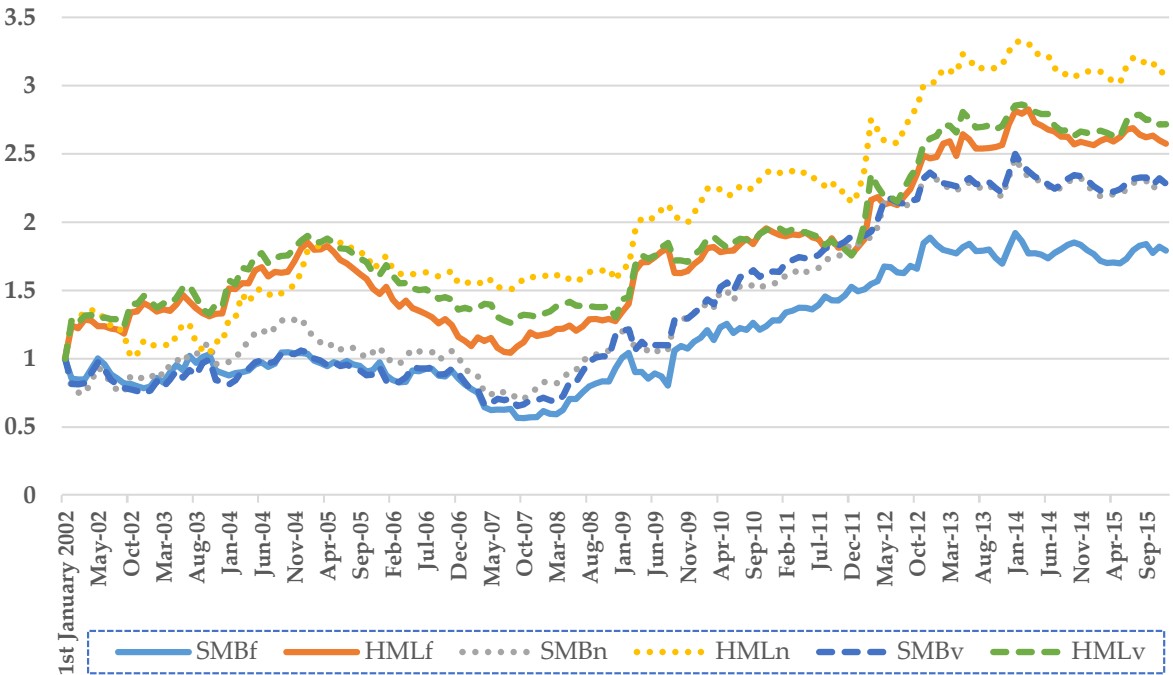

**Figure 1.** Cumulative value of the size and value factors using three different methodologies. Source: Author's own plotting. The sample period is 2002:01–2015:12. Source: the official website of the Pakistan stock exchange (https://www.psx.com.pk/) and the official website of the State Bank of Pakistan (http://sbp.org.pk/).

Overall, the results reveal that the three-factor model based on each of the basket of stocks explains the time-series variation in Pakistani stock returns very well. The fixed basket, used by most of the previous studies on the Pakistani stock market, performs the worst in terms of explanatory power and significance of the risk factors. The explanatory power of the three-factor model is relatively high when the portfolios include both financial and non-financial stocks (variable basket) as compared to when the portfolios include only non-financial stocks. Based on the explanatory power of the model, Sharpe ratio, cumulative wealth, and significance of the intercepts and the risk factors, we consider size and value factors constructed by using a variable basket of stocks for further analysis and robustness checks.

*4.4. Robustness Test*

As discussed earlier in this paper, our sample period includes the Global financial crises (2007–2009). Therefore, we break the sample into three sub-periods based on a combination of global and domestic market conditions, to confirm that our findings are robust. From January 2002 to December 2006 (pre-crises), from January 2007 to December 2010 (crises period) and from January 2011 to December 2015 (post-crises). Table A2 (Panel A, Panel B, and Panel C) in the appendix examines the time varying behavior of the three-factor model and size and book-to-market factors.

The $R^2$ of the six size-B/M portfolios' regressions in the post-crises period, with an average of approximately 81.82%, is higher than the other sub-periods, followed by crises (79.12%) and pre-crises (65.1%) periods, respectively. In the first sub-period (pre-crises), two out of six portfolios have statistically significant intercepts, whereas, in the second sub-period (crises), one portfolio has significant intercept, and in the third sub-period (post-crises), all the portfolios have insignificant intercepts. However, the magnitude of the intercepts is very nominal, ranging from 0.0001 to 0.0153 in the first sub-period, from −0.0143 to −0.0014 in the second sub-period, and from −0.0038 to 0.0039 in the third sub-period. All the coefficients on market factor across all the three sub-periods are significant at 1% level. With regard to size factor, the six size-B/M portfolios across the sub-periods exhibit varying degrees of sensitivity to the size factor, SMB. However, generally, the small portfolios have a large and positive sensitivity to SMB, whereas big portfolios have a small and negative sensitivity in first two sub-periods, and a nominal but positive sensitivity in the last sub-period. The average coefficients on small portfolios in each sub-period; pre-crises, crises and post-crises, are comparatively higher (0.762, 0.933 and 1.788, respectively) than the average coefficients on big portfolios (0.238, −0.067 and 0.288, respectively).

Our results show that size premium is getting stronger over the time period in terms of both the coefficients and the significance. Finally, value factor across the three sub-periods; pre-crises, crises and post-crises, high B/M ratio portfolios have a positive and large sensitivity to HML (0.908, 0.267 and 0.780, respectively), while low B/M stocks have a small and negative sensitivity to HML (−0.092, −0.733 and −0.220, respectively). The significance of the value factor is the highest in the post-crises period with approximately similar magnitude as in the pre-crises period. Our results for value factor confirm the findings of Davis et al. (2000), where significance of value premium increased from ($t$ = 2.8) to ($t$ = 3.38) in the recent sub-period. The value premium in our study is getting stronger over the time period. This is in contrast to the finding of Chung et al. (2016), where it was concluded that value premium is getting weaker over the time period in the Australian market.

It has been discussed that the data from the low-risk state are consistent with CAPM, whereas data from the high-risk state are inconsistent with CAPM (Huang 2000). The sample was divided into two different regimes (low risk and high risk). The results suggested that the data from the high risk regime violate CAPM. However, the data from the low risk state are consistent with CAPM. First, we regressed all the stocks on the market beta and classified them into 6 risk-based portfolios (B1 (high), B2, B3, B4, B5 and B6 (low)) to measure whether size and value premiums exist in all risk regimes. B1 represents stocks of the highest risk category (market beta), whereas B6 represents stocks at the lowest risk level. All portfolios contain an equal number of stocks.

Table 10 represents the estimation results of the three-factor model within a time-series context for each of the six risk-based portfolios. The regression results show that the coefficients of the market, size and value premiums are positive and significant at 1% level. The intercept is statistically insignificant for all the six portfolios and the $R^2$ values ranges between 36.56% and 74.68%. Our results support the existence and significance of the size and value premiums across all the risk profiles (regimes). However, the medium-ranked portfolios (B2, B3 and B4) have a higher explanatory power, and a higher level of significance for loadings on SMB and HML, with somewhat similar magnitude.

**Table 10.** Three factor regression on monthly excess returns of portfolios formed on risk profile (market beta).

| $R_i - R_f$ | $\alpha$ | $\beta$ | s | h | $R^2$ | Adj. $R^2$ |
|---|---|---|---|---|---|---|
| B1 (High) | 0.004 (0.697) | 1.286 (19.073) * | 0.644 (5.227) * | 0.206 (2.132) ** | 0.606 | 0.599 |
| B2 | −0.004 (−0.976) | 1.147 (18.865) * | 0.528 (6.435) * | 0.259 (4.026) * | 0.747 | 0.742 |
| B3 | −0.003 (−0.900) | 0.924 (17.903) * | 0.545 (7.827) * | 0.250 (4.585) * | 0.730 | 0.725 |
| B4 | −0.002 (−0.539) | 0.784 (14.738) * | 0.460 (6.411) * | 0.307 (5.453) * | 0.673 | 0.667 |
| B5 | 0.002 (0.387) | 0.641 (10.930) * | 0.559 (7.066) * | 0.264 (4.261) * | 0.536 | 0.528 |
| B6 (Low) | 0.005 (1.112) | 0.454 (7.329) * | 0.507 (6.070) * | 0.213 (3.244) * | 0.366 | 0.354 |

Note: Author's calculation. The table reports the estimation results of the three-factor model. Stocks are sorted into six risk-based portfolios. B1 contains securities of the highest risk level (highest market beta) whereas B6 contains the lowest risky securities. *t*-stats are in parenthesis, ** and * indicate significance at 5% and 1% level, respectively. The sample period is 2002:01–2015:12 (168 monthly observations). Source: the official website of the Pakistan stock exchange (https://www.psx.com.pk/) and the official website of the State Bank of Pakistan (http://sbp.org.pk/).

Petkova (2006) noticed a moderate explanatory power of the Fama–French factors on stock returns in the presence of macroeconomic risk factors. Elgammal et al. (2016) investigated the relationship between default premium and size and value premiums in the US market. They suggested that the default premium has explanatory power for value and size premiums. Baek and Bilson (2015) confirmed the existence of size and value premiums in both financial and nonfinancial firms. Additionally, they clarified that the financial firms are also sensitive to interest rate risk premium. Since our sample also includes financial firms, we examine the augmented four-factor model by including term structure premium into the Fama French three-factor model. The term structure premium (TSP) is calculated by finding the difference between the cut-off yield on ten-year Pakistan investment bonds (PIBs) and three-month Pakistani Treasury bill rate. By introducing TSP into the model, the relationship between excess returns and risk factors is modelled as:

$$E(R_i) - R_f = \alpha_i + \beta_i \left[ E(R_m) - R_f \right] + s_i(SMB) + h_i(HML) + ts_i(TSP) + \epsilon_i \tag{5}$$

where, $E(R_i) - R_f$ is the portfolio $i's$ return in excess of risk-free rate $R_f$, $\alpha_i$ is the intercept of the regression equation representing the non-market return component, $E(R_m) - R_f$ is the market risk premium (market portfolio return in excess of risk-free rate), SMB (small minus big) is the return on small size stocks minus return on big size stocks captures size premium, HML (high minus low) incorporates value premium that is the difference between returns of value stocks (high B/M ratio) and growth stocks (low B/M ratio), and $TSP$ (term structure premium) is calculated by finding the difference between the cut-off yield on ten-year PIBs and three-month T-bills of Pakistan. $\beta_i$, $s_i$, $h_i$, and $ts_i$ are the slopes of expected risk premium of portfolio $i$ to the market factor, size factor, value factors and term structure premium in the regression, respectively, while $\epsilon_i$ represents the random return component due to unexpected events related to a particular portfolio.

Results reported in Table 11 show that there is a negligible increase in the average adjusted $R^2$ due to the addition of TSP (from 71.23% to 71.73%). In contrast, there is a huge increase in the significance of the average intercept (from one portfolio to three portfolios) and magnitude of the average intercept (from 0.004 to 0.008). Our results demonstrate that SMB, HML and market factors remain robust to the inclusion of the term structure premium. Our findings for the three-factor model are robust across various portfolio construction techniques.

**Table 11.** Augmented four-factor regression on monthly excess returns of portfolios formed on size and B/M ratio (variable basket).

| $R_i - R_f$ | $\alpha$ | $\beta$ | s | h | ts | $R^2$ | Adj. $R^2$ |
|---|---|---|---|---|---|---|---|
| SL | −0.014 (−2.402) * | 0.970 (14.544) * | 1.186 (13.331) * | −0.294 (−4.220) * | 3.735 (1.818) *** | 0.639 | 0.630 |
| SM | (−0.009 (−2.066) ** | 0.744 (14.697) * | 0.741 (10.972) * | 0.202 (3.817) * | 3.707 (2.377) ** | 0.674 | 0.666 |
| SH | −0.001 (−0.230) | 0.894 (17.496) * | 1.189 (17.454) * | 0.794 (14.856) * | 2.581 (1.640) | 0.840 | 0.837 |
| BL | −0.003 (−0.599) | 0.845 (17.612) * | 0.028 (0.434) | −0.186 (−3.695) * | 2.978 (2.013) ** | 0.715 | 0.708 |
| BM | −0.006 (−1.224) | 0.840 (14.606) * | 0.064 (0.833) | 0.161 (2.680) * | 2.914 (1.643) *** | 0.682 | 0.675 |
| BH | −0.016 (−2.836) * | 0.921 (14.745) * | 0.024 (0.291) | 0.726 (11.102) * | 4.131 (2.146) ** | 0.793 | 0.788 |

Note: Author's calculation. The table reports the estimation results of an augmented Fama-French four-factor model that includes term structure premium. Stocks are sorted into six size-B/M portfolios (SL, SM, SH, BL, BM and BH). Term structure premium is calculated by finding the difference between the cut-off yield on ten-year PIBs and three-month T-bill of Pakistan. *t*-stats are in parenthesis, ***, ** and * indicate significance at 10%, 5% and 1% level, respectively. The sample period is 2002:01–2015:12 (168 monthly observations). Source: the official website of the Pakistan stock exchange (https://www.psx.com.pk/) and the official website of the State Bank of Pakistan (http://sbp.org.pk/).

## 5. Predictive Ability of the Three Factors for Future Economic Growth

Fama and French (1992, 1993, 1996, 1998) argue that SMB and HML act as state variables that predict future variations in the investment opportunities established in the context of intertemporal capital asset pricing model (Merton 1973). Liew and Vassalou (2000) attempt to link the return-based factors with future growth in the macro-economy. They conclude that HML and SMB contain significant information about future GDP growth, and risk-based explanation for the returns of SMB and HML is plausible. The evidence will enhance our understanding of whether or not these factors relate to underlying economic risk factors.

In this section, we discuss the third main objective of the study. This objective is to examine the relevance of the market factor, SMB, and HML with future GDP growth of Pakistan using univariate and multivariate regression analysis. Along with these factors, we include Treasury bill rate and term structure premium to predict Pakistan's GDP growth one year ahead. That is, we explore the ability of these factors at year $Y_{t-1}$ to forecast the GDP growth for year $Y_t$. The annual GDP data is obtained from the official website of the Asian development bank, whereas the data for T-bills and PIBs are obtained from the official website of the State Bank of Pakistan. Term structure premium is calculated by finding the difference between the cut-off yield on ten-year Pakistan investment bonds and three-month Pakistani Treasury bill rate, while GDP growth is calculated as the continuously compounded growth rate in Pakistan's gross domestic product. To obtain the yearly values, we have calculated the average (mean) of the monthly market risk premium (MKT), SMB, HML, Treasury bill rate (TB) and term structure premium (TSP) within each year (12 months).[8] The following equation represents the model:

$$GDP_{g,t} = \alpha + \beta\left[E(R_{m,t-1}) - R_{f,t-1}\right] + sSMB_{t-1} + hHML_{t-1} + fBCV_{t-1} + \epsilon_t \tag{6}$$

---

[8] See Boamah (2015) and Liew and Vassalou (2000) for an extensive overview of the methodology.

where, $GDP_{g,t}$ represents the GDP growth at time $t$ and $BCV_{t-1}$ represents the business cycle variables, which refers to the term structure premium and the three-month Treasury bill rate.

Before we proceed with the main test, we examine the stationarity of the variables. The market factor and HML are stationary, while SMB, GDP growth, T-bill and term structure premium are taken as first-order difference. The absence of a unit root in the series of returns is confirmed by augmented Dickey-Fuller test and Phillips-Perron test, with trend and intercept. It is carefully observed that all the variables are stationary at the time we perform regressions. Next, we examine various versions of Equation (6) and present the results in Table 12. To check whether the remaining residuals are independent and identically distributed (i.i.d.), we have conducted the BDS test by Broock et al. (1996) and no nonlinearity is found.

The evidence shows that in univariate regressions, the market factor and size premium show significant association with future growth of the Pakistani GDP, whereas value premium, Treasury bill and term structure premium indicate an insignificant relationship with future growth in GDP.

It is further evident that only the coefficient of the market factor is positive, whereas the coefficients of SMB, HML, TB and TSP are negative. The explanatory power of the univariate regressions is 63.03%, 32.79%, 1.30%, 9.32% and 1.02% for the market, SMB, HML, TB and TSP, respectively. The findings indicate that the predictive ability of the market factor and SMB for the growth of the Pakistan's GDP is non-trivial.

In a two-factor model consisting of market factor and SMB, HML, TB and TSP, the result indicates that the coefficients of market factor are all positive and statistically significant. The loadings on HML, SMB, Treasury bill and term structure are negative, but insignificant. The $R^2$ values are relatively higher in the two-factor regression, ranging from 63.32% (market and TSP) to 68.86% (market and TB). The two-factor model consisting of SMB and HML indicates that including HML into the regression model does not subsume the significance of SMB factor. The negative coefficients of SMB are similar to the findings of Liew and Vassalou (2000) for Switzerland and Japan.

**Table 12.** The information content of market, SMB and HML for future economic growth.

| Model | $\alpha$ | MKT | SMB | HML | TB | TSP | $R^2$ |
|---|---|---|---|---|---|---|---|
| 1 | 0.095 | 0.170 * | | | | | 0.630 |
| 2 | 0.040 | | −0.217 ** | | | | 0.328 |
| 3 | 0.015 | | | −0.003 | | | 0.013 |
| 4 | 0.018 | | | | −1.533 | | 0.093 |
| 5 | 0.013 | | | | | −0.647 | 0.010 |
| 6 | 0.093 | 0.155 * | −0.041 | | | | 0.637 |
| 7 | 0.092 | 0.183 * | | −0.040 | | | 0.659 |
| 8 | 0.100 | 0.166 * | | | −1.217 | | 0.689 |
| 9 | 0.083 | 0.169 * | | | | −0.342 | 0.633 |
| 10 | 0.011 | | −0.241 ** | | −2.566 ** | −1.453 | 0.543 |
| 11 | 0.034 | | | 0.006 | −2.046 | −1.687 | 0.154 |
| 12 | 0.062 | | −0.250 ** | 0.068 | | | 0.412 |
| 13 | 0.075 | | −0.272 * | 0.058 | −1.988 | | 0.562 |
| 14 | 0.065 | | −0.250 ** | 0.068 | | 0.079 | 0.412 |
| 15 | 0.092 | 0.200 ** | 0.035 | −0.052 | | | 0.662 |
| 16 | 0.096 | 0.170 ** | 0.023 | −0.041 | −1.397 | | 0.730 |
| 17 | 0.073 | 0.206 ** | 0.049 | −0.058 | | −0.541 | 0.669 |
| 18 | 0.036 | | −0.265 * | 0.051 | −2.361 *** | −1.156 | 0.588 |
| 19 | 0.049 | 0.177 * | −0.004 | −0.054 | −1.829 | −1.411 | 0.769 |

Note: Author's calculation. The table reports the estimation results of the Fama-French three-factor model to predict future GDP growth of Pakistan. The MKT, SMB, HML and BC are correspondingly the excess return to the market risk premium, size premium, value premium, and business cycle variables (Treasury bill (TB) and term structure premium (TSP)). *t*-stats are in parenthesis, ***, ** and * are the 10%, 5% and 1% significance level, respectively. The sample period is 2002–2016 (168 monthly observations of risk factors (converted into annual values), and 14 annual observations of GDP growth). Source: the official website of the Pakistan stock exchange (https://www.psx.com.pk/), the official website of the State Bank of Pakistan (http://sbp.org.pk/) and the Asian development bank (https://www.adb.org/data/south-asia-economy).

In a multivariate regression analysis, Table 12 further reports that including Treasury bill, term structure premium or both in the model does not subsume the relevance of the SMB. The market factor has the strongest relevance with the GDP and deteriorates all the other factors. However, inclusion of HML and business cycle variables (TB and TSP) does not eliminate the forecasting of SMB. The evidence in this study suggests that the market factor and SMB possess the information content for one year ahead Pakistan's GDP growth. The negative relation of SMB with future economic growth, presumably, indicates that the investors would rather hold the big capitalization stocks when they notice that the economy is in bad state (low or instable growth).

## 6. Conclusions

Using monthly data from Pakistan's Karachi stock exchange (KSE) between 2002 and 2016, the article conducts an empirical investigation of the Fama-French's three-factor model. Specifically, this article inspects three different ways (fixed basket, non-financial basket, and variable basket) of constructing size and value factors in order to gauge the effects of the special features in Pakistani stock market. Our main findings are as follows.

First, the findings demonstrate that the formation of the Fama-French factors can have a significant impact in empirical studies that apply the Fama-French models to Pakistani stock returns. We recommend that the risk factors be constructed by including both financial and non-financial companies, where the model explains about 71.23% of the variations in the stock returns on Pakistani market. It is noticed that the average $R^2$ values of the three-factor model are meaningfully higher than those of the CAPM. Since our sample includes financial firms, an augmented four-factor model that includes term structure premium (TSP) is tested. Although the loadings on TSP are mostly significantly positive, but the relevance of size (SMB), value (HML) and market factors is not deteriorated. The four-factor model does not improve the explanatory power of the model, whilst it increases the significance and the magnitude of the average intercept.

Second, the study explores the ability of the SMB, HML and market factors to predict future growth of the Pakistani economy (GDP). The paper provides evidence of statistically significant and positive relation between future growth of the Pakistani economy and the market factor, which is robust in the presence of SMB, HML and the business cycle variables in the models. Further evidence shows negative and statistically significant relationship between future growth of the Pakistani economy and SMB, whilst the loadings on the HML, T-bill and term structure premium are negative, but statistically insignificant. The market and size factors are robust to the inclusion of the business cycle variables in the model. The negative relation of SMB with future economic growth, presumably, indicate that the investors would rather hold the big capitalization stocks when they notice that the economy is in a bad state (low or instable growth).

Third, the robustness test confirms that the three-factor model captures the time-series variations in stock returns across the three sub-periods (pre-, during-, and post-crises), six risk regimes (portfolios' risk profile), and across three different portfolio construction methodologies (baskets of stocks). However, the significance and coefficients vary over time, across risk-profile of the portfolio, and across portfolio construction methodology. The three-factor model performs better in the post-crises period (2010–2015): (1) the average value for $R^2$ is the highest, approximately 81.82%; (2) intercepts are statistically insignificant for all the six LHS portfolios; and (3) loadings on the market, SMB and HML factors are mostly significant at 1% level.

By and large, considering the empirical evidence, across the different estimation techniques and methodologies, we find that the size and book-to-market (value) are the factors significantly and consistently exist in the Pakistani equity returns; however, the significance and magnitude of these factors and the three-factor model vary. Most importantly, these factors, except for HML, have relevance with the future growth of the Pakistani economy. Being a small open economy, factors such as foreign investors trading (Ceylan et al. 2015) may have influence on the stock prices and future

economic growth of Pakistan. The study of these factors will be worth doing in the future to further understand special characteristics of KSE, Pakistan.

**Author Contributions:** Fahad Ali and RongRong He contributed to data collection and management, and interpreted the results; YueXiang Jiang contributed to the analysis of the estimation results; Fahad Ali and YueXiang Jiang provided analytical materials and methodological tools; Fahad Ali, RongRong He and YueXiang Jiang wrote the manuscript. All authors read and approved the final manuscript.

**Conflicts of Interest:** The authors declare no conflict of interest.

## Appendix A

**Table A1.** CAPM regression on monthly excess returns of portfolios formed on size and B/M ratio (fixed basket and non-financial basket).

| $R_i - R_f$ | $\alpha$ | $\beta$ | $R^2$ | Adj. $R^2$ |
|---|---|---|---|---|
| **Panel A: Fixed basket** | | | | |
| SL | 0.007 (0.985) | 0.503 (5.510) * | 0.154 | 0.150 |
| SM | 0.008 (1.844) ** | 0.538 (9.367) * | 0.346 | 0.342 |
| SH | 0.021 (3.177) * | 0.727 (8.338) * | 0.597 | 0.594 |
| BL | 0.005 (1.372) | 0.739 (15.667) * | 0.597 | 0.594 |
| BM | 0.002 (0.343) | 0.881 (15.481) * | 0.642 | 0.640 |
| BH | 0.003 (0.567) | 0.940 (14.491) * | 0.558 | 0.556 |
| **Panel B: Non-financial basket** | | | | |
| SL | 0.009 (1.117) | 0.661 (6.625) * | 0.209 | 0.204 |
| SM | 0.009 (2.017) * | 0.549 (9.074) * | 0.332 | 0.328 |
| SH | 0.028 (3.770) * | 0.792 (8.382) * | 0.297 | 0.293 |
| BL | 0.006 (1.541) | 0.691 (14.150) * | 0.547 | 0.544 |
| BM | 0.002 (0.573) | 0.916 (17.188) * | 0.640 | 0.638 |
| BH | 0.003 (0.506) | 1.154 (15.301) * | 0.585 | 0.583 |

Note: Author's calculation. The table reports the estimation results of the CAPM (fixed and non-financial basket). Stocks are sorted into six size-B/M portfolios (SL, SM, SH, BL, BM, BH). *t*-stats are in parenthesis, ** and * indicate significance at 5% and 1% level, respectively. The sample period is 2002:01–2015:12 (168 monthly observations). Source: the official website of the Pakistan stock exchange (https://www.psx.com.pk/) and the official website of the State Bank of Pakistan (http://sbp.org.pk/).

**Table A2.** Three factor regression on monthly excess returns of portfolios formed on size and B/M ratio (subperiods).

| $R_i - R_f$ | $\alpha$ | $\beta$ | s | h | $R^2$ | Adj. $R^2$ |
|---|---|---|---|---|---|---|
| **Panel A: January 2002 to December 2006 (pre-crises)** | | | | | | |
| SL | 0.002 (0.207) | 0.796 (6.462) * | 0.815 (4.369) * | −0.114 (−0.838) | 0.449 | 0.420 |
| SM | 0.004 (0.687) | 0.638 (7.145) * | 0.652 (4.822) * | 0.401 (4.057) * | 0.637 | 0.618 |
| SH | 0.015 (2.226) ** | 0.695 (7.315) * | 0.819 (5.690) * | 0.852 (8.112) * | 0.763 | 0.751 |
| BL | 0.014 (2.120) ** | 0.711 (8.054) * | −0.160 (−1.197) | −0.069 (−0.708) | 0.658 | 0.639 |
| BM | 0.008 (0.941) | 0.607 (5.180) * | −0.391 (−2.203) ** | 0.244 (1.883) *** | 0.601 | 0.579 |
| BH | 0.000 (0.010) | 0.812 (6.984) * | −0.164 (−0.930) | 0.964 (7.502) * | 0.802 | 0.791 |
| **Panel B: January 2007 to December 2010 (crises period)** | | | | | | |
| SL | −0.014 (−1.609) | 1.001 (10.077) * | 1.124 (6.863) * | −0.831 (−4.979) * | 0.769 | 0.754 |
| SM | −0.006 (−0.951) | 0.804 (10.204) * | 0.667 (5.131) * | −0.182 (−1.372) | 0.705 | 0.685 |
| SH | −0.001 (−0.233) | 0.895 (13.075) * | 1.009 (8.935) * | 0.357 (3.106) * | 0.816 | 0.803 |
| BL | −0.002 (−0.379) | 0.836 (12.792) * | −0.207 (−1.919) *** | −0.635 (−5.781) * | 0.841 | 0.830 |
| BM | −0.005 (−0.755) | 0.923 (12.171) * | 0.097 (0.776) | −0.198 (−1.553) | 0.805 | 0.792 |
| BH | −0.014 (−1.904) *** | 0.942 (10.630) * | −0.091 (−0.626) | 0.177 (1.191) | 0.811 | 0.798 |
| **Panel C: January 2011 to December 2015 (post-crises)** | | | | | | |
| SL | −0.004 (−0.727) | 1.152 (10.336) * | 1.345 (12.522) * | −0.279 (−3.578) * | 0.799 | 0.788 |
| SM | 0.001 (0.185) | 0.799 (7.805) * | 0.766 (7.755) * | 0.206 (2.876) * | 0.747 | 0.733 |
| SH | 0.003 (0.925) | 1.128 (14.975) * | 1.465 (20.171) * | 0.825 (15.662) * | 0.959 | 0.957 |
| BL | 0.004 (1.019) | 0.987 (11.919) * | 0.224 (2.809) * | −0.161 (−2.775) * | 0.727 | 0.713 |
| BM | −0.001 (−0.132) | 1.080 (13.473) * | 0.246 (3.186) * | 0.177 (3.149) * | 0.836 | 0.827 |
| BH | −0.003 (−0.586) | 1.011 (8.935) * | 0.105 (0.963) | 0.736 (9.299) * | 0.841 | 0.833 |

Note: Author's calculation. The table reports the estimation results of the three-factor model. Stocks are sorted into six size-B/M portfolios (SL, SM, SH, BL, BM, BH). *t*-stats are in parenthesis, ***, ** and * indicate significance at 10%, 5% and 1% level, respectively. The sample period is 2002:01–2015:12 (168 monthly observations). Source: the official website of the Pakistan stock exchange (https://www.psx.com.pk/) and the official website of the State Bank of Pakistan (http://sbp.org.pk/).

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
