# Peer review of "Size, Value and Business Cycle Variables. The Three-Factor Model and Future Economic Growth: Evidence from an Emerging Market"

_economies, doi:10.3390/economies6010014_

Round 1

Reviewer 1 Report

I have the following comments for the improvement of the paper;

Abstract is too long. Abstract is to communicate the one central and novel contribution of your paper. So 150 to 200 words enough.

Author need to be consistent with data range, like in introduction page 3 it is mentioned that data is from 2002 to 2016. However in other parts of paper it's 2012 to 2015.Why there is one year difference?

In the data section, it is said that author used only active firms, what about  dead firms. If the author is using only active firms what about survivorship bias?

Is there any specific reason of using 6 month T bills rate? As you are using all other data on monthly frequency, why not T.bills on monthly basis.

Author Response

Dear Editor and Referee

We are thankful to you for your valuable comments and suggestions to improve the manuscript. Please find below point-by-point response to each of the reviewer’s comments. The changes made in this version of manuscript are highlighted in yellow.

Moreover, we have read the manuscript carefully and all mistakes related to language have now been corrected.

Once again, thank you for your time and consideration.

Best Regards,

Abstract is too long. Abstract is to communicate the one central and novel contribution of your paper. So 150 to 200 words enough.

Response: Abstract is reduced to 173 words, as suggested (L5-L16, Page1).

Author need to be consistent with data range, like in introduction page 3 it is mentioned that data is from 2002 to 2016. However in other parts of paper it's 2012 to 2015.Why there is one year difference?

Response: The overall dataset is from 2002 to 2016. The earlier portion of the paper discusses the three-factor model between 2002 and 2015, whereas, the later portion examines the relationship between these risk factors to predict the future economic growth between 2003 and 2016. To make it clear in the introduction section, as it describes the earlier portion, we have now changed it to 2002-2015 (L111, Page 3).

In the data section, it is said that author used only active firms, what about dead firms. If the author is using only active firms what about survivorship bias?

Response: We carefully examined the stocks of approximately 650 firms. However, 350 firms were finally chosen based on sample selection criteria and limitations, as described in the section 3.2. We particularly focused on firms which had been traded for at least 85% of the trading days during the year. Generally, if we had included the dead firms, the liquidity condition of the ‘selection criteria and limitations’, mentioned in the section 3.2 of the paper, would not have met.

Actively traded firms refer to those which remained traded during the year. Dead firms were basically excluded due to illiquidity (zero returns), however, if a dead firm remained liquid during any specific year, it was considered in that particular year. This matter (survivorship bias) is further explained and now highlighted in sections 3.1 (L284-L285, Page 7) and 3.4 (L321-L323, Page 8).

Is there any specific reason of using 6 month T bills rate? As you are using all other data on monthly frequency, why not T.bills on monthly basis.

Response: Yes, the previous studies on Pakistan (Iqbal and Brooks 2007); (Mirza and Shahid 2008) have considered 6 month T.bills rate. So we followed the same practice. Now these two references are highlighted to support our idea (L280-L281, Page 7).

Reviewer 2 Report

This is a very interesting paper that provides an extensive research into the applicability of Fama-French factors to the main stock market index in Pakistna, KSE. I found the research sound and well executed. 

I suggest a few robustness tests:

- running the key regressions before and after the financial crisis; how does this affect the results?

- considering daily data for at least a small subset of regressions in the paper.

I also suggest discussing  whether Pakistan, as a small open economy, is affected in any way by factors related to small open economies, even though the standard framework on factor models does not take into account this. Some studies addressed this:

http://www.tandfonline.com/doi/pdf/10.1080/1331677X.2015.1075138

Author Response

Dear Editor and Referee

We are thankful to you for your valuable comments and suggestions to improve the manuscript. Please find below point-by-point response to each of the reviewer’s comments. The changes made in this version of manuscript are highlighted in yellow.

Moreover, we have read the manuscript carefully and all mistakes related to language have now been corrected.

Once again, thank you for your time and consideration.

Best Regards,

I suggest a few robustness tests:

- running the key regressions before and after the financial crisis; how does this affect the results?

Response: Yes, the robustness checks were done, and the results are highlighted now. The result shows the R2 of the post-crises period is approx. 81.82%, while it was approx. 65.15% in the pre-crises period (L519-L521). Furthermore, the Size and value premiums are stronger in the post-crises period than the pre-crises period, (L534-L535, Page 16) and (L541-542, Page 16). Most importantly, all the intercepts are statistically insignificant in the post-crises period, whereas in the pre-crises period two portfolios out of four produce significantly positive intercepts (L521-L524, Page 16).

- considering daily data for at least a small subset of regressions in the paper.

Response: Davis (1994) reported, the frequency of data does not improve or deteriorate results. Moreover, it is a common practice to use monthly returns in the relevant literature related to multi-factor models and the prediction of future economic growth, such as Fama-French (1992, 1993, 1996, 2015, and 2017) and many other studies which extended similar kind of work.

However, several other robustness checks were performed and presented in the paper, such as robustness across subperiods, risk regimes, and different methodologies to form factors. In addition, robustness check were also performed to the inclusion of business cycle variables.

A few other robustness tests were performed, but were not included in the paper in order to avoid it to become too lengthy. These robustness checks include calendar effects and checking validity of the risk factors at the individual stock level (instead of making portfolios).

I also suggest discussing whether Pakistan, as a small open economy, is affected in any way by factors related to small open economies, even though the standard framework on factor models does not take into account this. Some studies addressed this:

http://www.tandfonline.com/doi/pdf/10.1080/1331677X.2015.1075138

Response: I believe it would be an interesting factor(s) to examine. However, studying the effects of these factors was beyond the scope of this study. Our work primarily focused on special characteristics of KSE, Pakistan, which were more relevant for the purpose of estimating size and value premiums, and the role of business cycle variables in the model. Yet, in the conclusion section we added and highlighted this recommendation for future study (L683-L686, Page 20).
